# What Is the Nutritional Quality of Pre-Packed Foods Marketed to Children in Food Stores? A Survey in Switzerland

**DOI:** 10.3390/nu16111656

**Published:** 2024-05-28

**Authors:** Fabien Pellegrino, Monique Tan, Celine Richonnet, Raphaël Reinert, Sophie Bucher Della Torre, Angeline Chatelan

**Affiliations:** 1Department of Nutrition and Dietetics, Geneva School of Health Sciences, HES-SO University of Applied Sciences and Arts Western Switzerland, Rue des Caroubiers 25, 1227 Carouge, Switzerland; 2Wolfson Institute of Preventive Medicine, Barts and the London School of Medicine and Dentistry, Queen Mary University of London, Charterhouse Square, London EC1M 6BQ, UK; 3Club Européen des Diététiciens de l’Enfance (CEDE), Esplanade, 17-7800 Ath, Belgium; 4Federal Food Safety and Veterinary Office, Schwarzenburgstrasse 155, 3003 Bern, Switzerland

**Keywords:** food marketing, children, nutritional quality, Nutri-Score, ultra-processed food, nutrient profiling, food packaging, food composition

## Abstract

Food marketing targeting children influences their choices and dietary habits, and mainly promotes food high in fat, sugar, and salt as well as ultra-processed food. The aim of this study was to assess the nutritional quality of food and beverages marketed to children over the age of 3 and available on the Swiss market. Products with at least one marketing technique targeting children on the packaging were selected from five food store chains. Three criteria to assess nutritional quality were used: (1) nutritional composition (using the Nutri-Score), (2) degree of processing (NOVA classification), and (3) compliance with the World Health Organization (WHO) Nutrient Profile Model (NPM). A total of 735 products were found and analyzed. The most common marketing techniques used were childish names/fonts (46.9%), special characters (39.6%), and children’s drawings (31.3%). Most products had a Nutri-Score of D or E (58.0%) and were ultra-processed (91.8%). Only 10.2% of products displayed the Nutri-Score. The least processed products generally had a better Nutri-Score (*p* < 0.001). Most products (92.8%) did not meet the criteria of the WHO NPM. Products that met the WHO NPM criteria, organic products, and products with a nutritional claim generally had a better Nutri-Score and were less processed (p_s_ < 0.05). Pre-packaged foods and beverages marketed to children in the Swiss market were mostly of poor nutritional quality. Public health measures should be adopted to improve the nutritional quality of foods marketed to children in Switzerland and restrict the marketing of unhealthy foods to children.

## 1. Introduction

The prevalence of overweight and obese children is a public health problem [1,2,3]. The World Health Organization (WHO) estimates that 39 million children under the age of 5 and 340 million children aged between 5 and 19 were overweight or obese in 2019 [2]. In Switzerland, 17.5% of children aged between 4 and 15 suffered from being overweight and 4.7% from obesity in 2022 [4]. Overweight and obese children have a higher risk of non-communicable diseases (e.g., cardiovascular disease and type 2 diabetes) [2,5,6] and premature death [2]. An unbalanced diet, characterized by the consumption of products that are high in fat, sugar, and/or salt (HFSS) and ultra-processed foods (UPFs), is one of the main causes of childhood obesity [6,7,8,9].

Worldwide, children are highly exposed to HFSS products and UPFs through advertising using child-oriented marketing techniques (e.g., brand mascots, toys, and cartoon characters) [10,11,12,13]. WHO recently reported that the proportion of food marketing to children promoting HFSS products was generally greater than 50% and up to 90% [10]. A recent study analyzing food marketing to children in 22 countries across the world revealed that unhealthy products were advertised up to four times more often than healthy ones [14]. On television, it has been demonstrated in several European countries that children see up to ten times more advertisements per hour for unhealthy food than for healthy food [15]. In Switzerland, a recent study found that 11.7% of advertisements shown to children on social networks were for foods and beverages, promoting mainly chocolate and sweets [16]. This exposure has a negative impact on children’s health [7,17,18] by influencing their food choices, preferences, and purchase requests and by contributing to unhealthy dietary habits [10,17,19,20,21]. Food marketing to children plays, therefore, an important role in promoting unhealthy foods, and, therefore, contributes to non-communicable diseases [7], including obesity [7,17,22]. 

The nutritional quality of food products marketed to children can be assessed in different ways, notably according to their nutritional composition or the degree of processing. Several studies in countries such as Canada, the United Kingdom, and Slovenia have analyzed the nutritional quality of food and beverages marketed to children and found that the majority of these products were of poor nutritional quality [11,12,13,23,24]. More specifically, in France, which is geographically close to Switzerland, Richonnet and colleagues found that 88% of the 1,152 pre-packaged foods or beverages marketed to children they found in 20 food store chains were ultra-processed and most contained flavorings and ultra-processed sugars [24]. 

In Switzerland, a voluntary initiative to restrict marketing to children is in place: the Swiss Pledge, signed by leading food and beverage companies [25], is guided by the restrictions prescribed by the EU Pledge in Europe [26]. The EU Pledge (and by extension the Swiss one) covers only digital marketing and not food packaging [25,26]. In addition, there is no study that has assessed the nutritional quality of foods marketed to children in Swiss stores yet. Therefore, the aim of this study was to analyze the nutritional quality of pre-packaged food marketed to children based on (i) the Nutri-Score labeling system as a proxy of nutritional composition [27], (ii) the degree of processing using the NOVA classification [28], and (iii) compliance with the WHO Regional Office for Europe Nutrient Profile Model (WHO NPM) [29]. Furthermore, we aimed to assess whether the nutritional quality of organic products or those with a nutrition claim was better than conventional products or those without a nutrition claim.

## 2. Materials and Methods

### 2.1. Selection of Food Products

The purpose was to reach a comprehensive sample of pre-packed foods marketed to children living in Switzerland. To do so, after receiving authorization from store managers, a registered dietitian (F.P.) visited five food stores from five different chains in the French-speaking regions of Switzerland: i.e., the two largest supermarkets (Migros and Coop), which held 64% of the retail market share in 2023 [30], two major hard discounters (Lidl and Aldi) and one grocery store selling only organic products (Kiss the Ground, one of the largest organic store in the French-speaking regions of Switzerland). The reason for including organic products is that (i) Switzerland is the country with the highest consumption of organic products in the world [31] and (ii) organic products are sold in every supermarket. This cross-sectional survey was conducted between 24 April and 3 May 2023. 

All food and beverages with at least one marketing technique targeting children aged 3 to 18 [29] on the packaging were included in this study. The marketing techniques could be as follows [10,11,12,13,14,23,24,32,33,34,35,36,37]: (i) children’s drawings (cartoons, animal drawings), (ii) cartoon characters, (iii) special characters (mascots created by the brand, personified food products), (iv) childish names/fonts (fun/animated fonts, special effects/colors), (v) games/contests on the packaging or incentives to play online games, (vi) gifts likely to be popular for children (prizes, physical or virtual gifts for children like toys, puzzles, activities), (vii) direct references to children (drawings/texts relating to children, use of words relating to childhood like “children”, use of the French informal “you”, mention that the product is intended for children, and portion size adapted to children), (viii) special packaging (animated, collectible, playful, and with attractive/unusual colors), (ix) reference to “fun” or a playful feeling when consuming these products, (x) reference to celebrities or popular athletes among children, and (xi) reference to children’s entertainment (movies, TV programs, games, and websites) [13,33,37]. Baby food products were not included because they are mainly purchased by parents, and they are targeted to an age group that has specific nutritional requirements [29]. Each included product had its packaging photographed. If the same product was found in different stores, it was recorded only once.

### 2.2. Definition of Nutritional Quality

The nutritional quality of products was assessed by a registered dietitian (F.P.) using (i) the Nutri-Score labeling system as a proxy of the quality of nutritional composition [27], (ii) the NOVA classification for the degree of processing [28], and (iii) the compliance to the WHO NPM for the conformity to be marketed to children [29].

The Nutri-Score is a front-of-pack nutrition label that can be displayed on prepacked foods by manufacturers on a voluntary basis. The Nutri-Score is associated with five letters (A to E) and corresponding five colors (dark green (healthiest) to dark orange (least healthy)) [27]. It accounts for the amount of nutrients or foods per 100 g/mL that should be limited (i.e., energy, saturated fatty acids, sugars, and salt) and encouraged (i.e., fibers, proteins, fruits, vegetables, legumes, nuts, and olive/rapeseed/walnut oils). The algorithm is different depending on the food category (general foods, cheeses, added fats, and beverages). All products were classified into one of the five letters of the Nutri-Score (A, B, C, D, or E) using the Nutri-Score calculation tool provided by Santé Publique France [27]. The Nutri-Score algorithm has been updated and the new calculation was adopted on 31 December 2023 with a two-year transition period [38]. In our study, the updated algorithm was not used because it was unavailable at the time of data analysis. 

The proportions of fruits, vegetables, nuts, legumes, and oils were extracted from the ingredient list. When nutritional values were missing on the packaging, data were extracted for a similar product using the Swiss Food Composition Database [39] or, if the product was not referenced there, from the French Food Composition Database Ciqual [40]. For products that did not mention the proportions of fruits, vegetables, nuts, legumes, and oils, the minimum proportion was deducted using the proportions mentioned for the other ingredients and their position in the ingredient list. If above 40%, the Nutri-Score was calculated with different thresholds (>40%, >60%, and >80%, corresponding to the different cut-offs when calculating the score). If the minimum proportion was not deductible due to lack of information and/or if the Nutri-Score score changed when calculated with the different thresholds, we conservatively assumed the highest proportion (>80%). 

The NOVA classification takes the physical, biological, and chemical techniques (including additives) used during food processing into account to classify the products into four groups: unprocessed or minimally processed foods (NOVA 1; e.g., fresh or dry fruit); processed culinary ingredients (NOVA 2; e.g., sugar and oil); processed food products (NOVA 3; e.g., cheese and canned vegetables); and ultra-processed food products (NOVA 4; soft drinks and breakfast cereals) [28]. The NOVA classification was assessed from the ingredient list, using Monteiro et al.’s description of the groups [28]. For instance, NOVA 4 products had a long list of ingredients, UPF markers (e.g., high-fructose corn syrup and hydrogenated fats), additives (e.g., thickeners and flavorings), and/or were made using highly processed industrial techniques (e.g., extrusion, molding and pre-frying). 

The WHO NPM was developed by the WHO Regional Office for Europe in 2015 [41] and updated in 2023 [29]. Its purpose is to classify foods according to their nutritional composition for nutrition policies aiming at preventing disease and promoting health, such as restricting the marketing of unhealthy foods to children [29]. Criteria compliance of the products was defined based on the nutritional thresholds set in 2023. The model has 22 categories (17 foods and 5 drinks). For each category, thresholds per 100 g/mL of product are set for the nutrients and other components (i.e., energy, total fat, saturated fat, trans fatty acids, total sugars, added sugars, non-sugar sweeteners, and sodium/salt). If the product exceeds any of the relevant thresholds for the product category, marketing is not permitted [29]. Compliance with the trans fatty acids criteria (1 g/100 g) could not be assessed because trans fatty acids content was not available on the packaging. Trans fatty acids content in products is limited to 2% by law in Switzerland [42]. 

### 2.3. Data Entry

For each food or beverage, information displayed on the packaging was collected following a standardized and piloted protocol and entered into an Excel database. The registered dietitian (F.P.) systematically recorded the (i) product name; (ii) used marketing technique(s) (11 pre-coded categories as described above); (iii) nutritional values per 100 g/mL for energy (kcal/kJ), protein, fat, saturated fatty acids, carbohydrates, sugars, fiber, sodium, and salt; (iv) proportion of fruits, vegetables, nuts, legumes, and olive/rapeseed/walnut oils; (v) presence of the Nutri-Score logo (yes/no); (vi) degree of food processing (NOVA 1—unprocessed, NOVA 2—processed culinary ingredients, NOVA 3—processed or NOVA 4—ultra-processed) and free-text justification for NOVA classification; (vii) food category according to the WHO NPM; (viii) conformity to WHO NPM; (ix) organic food (yes/no); (x) presence of nutrition or health claim(s) (yes/no) and type of claims (free-text summary); (xi) presence of free sugars (yes/no) according to the WHO definition [43]; and (xii) added salt (yes/no). 

### 2.4. Data Quality Check

To check the validity of the data entry, a second senior registered dietitian (A.C.) assessed 10 randomly selected products. The data were mostly in agreement: 90% for the calculated Nutri-Score (one Nutri-Score category away: B to C due to data entry error for a nutritional value), 100% for the NOVA classification, 100% for the presence of marketing, 30% for the marketing techniques used (A.C. confused techniques i and iii), 80% for the WHO NPM category, 100% for the compliance with the WHO NPM, 80% for the nutrition claims (data entry omission and A.C. considered “palm oil free” as a claim) and 100% for the organic label. The differences were discussed until a consensus was reached. Overall, quality control showed that data collection was reproducible when used by different dietitians.

### 2.5. Statistical Analysis

Frequencies (n) and relative frequencies (%) were calculated for categorical variables. Means and standard deviation (SD) were calculated for numeric variables. To measure the association between categorical variables, we used three tests: (i) the Chi-squared (χ^2^) test (NOVA classification and compliance with WHO NPM, presence of organic label, and compliance with WHO NPM, nutrition claims and compliance with WHO NPM, NOVA categories and organic products, NOVA categories and nutrition claims), (ii) the Wilcoxon-Mann-Whitney rank sum test (Nutri-Score and compliance with WHO NPM, Nutri-Score and organic products, Nutri-Score and nutrition claims), and (iii) the Kruskal-Wallis test (Nutri-Score and NOVA categories). All statistical analyses were performed using STATA software, version 17 (Stata Corporation, College Station, TX, USA). A *p*-value of <0.05 was considered statistically significant.

## 3. Results

### 3.1. Characteristics of the Sample

A total of 735 food products marketed to children aged 3 to 18 years were found in the five food stores and analyzed. For 5 products (0.7%), mainly herbal tea bags and bottled water, no nutritional value was indicated on the packaging. For 2 products (0.3%), ice cream and cream, saturated fat, sugars, and dietary fiber content were not mentioned. For 363 products (49.4%), dietary fiber content was not indicated. These nutritional values were deducted from the Swiss or French food composition tables for 296 products (80.0%), and 74 products (20.0%), respectively. For 20 products (2.7% of the total sample), information on the proportions of fruits, vegetables, nuts, legumes, and oils was missing, and their minimum deducted proportion exceeded 40%. This concerned mixed nuts (*n* = 6), bars with nuts and dried fruits (*n* = 6), mixed dried fruits (*n* = 2), and muesli (*n* = 1). For 17 products, the calculated Nutri-Score did not change, either with the minimum (>40%) or maximum (>80%) proportion assigned. For 3 products (0.4% of the total sample), mainly mixed nuts, the calculated Nutri-Score changed, so the highest proportion of nuts (>80%) was attributed to these products. 

The most common WHO NPM categories were confectionaries and desserts (*n* = 182, 24.8%); savory snacks (*n* = 122, 16.6%); and cakes, sweet biscuits, and pastries (*n* = 93, 12.7%). No products belonged to butter, other fats and oils, fresh and frozen meat, poultry and fish, fresh and frozen fruit, or vegetables and legumes. The distribution of products by category is detailed in Appendix A. Out of the 735 food products, 608 (82.7%) contained free sugars and 350 (47.6%) contained added salt.

### 3.2. Marketing Techniques

Between one and six marketing techniques were found on the packaging, representing a mean of 2.0 (±1.1) marketing techniques per product. The most common marketing techniques used were childish names/fonts (*n* = 345, 46.9%), special characters (*n* = 291, 39.6%), and children’s drawings (*n* = 230, 31.3%). The distribution of marketing techniques is shown in Figure 1.

### 3.3. Nutri-Score

In our sample (*n* = 735), following our calculation of the Nutri-Score, 55 products (7.5%) had a Nutri-Score of A, 90 products (12.2%) had a Nutri-Score of B, 164 products (22.3%) had a Nutri-Score of C, 288 products (39.2%) had a Nutri-Score of D, and 138 products (18.8%) had a Nutri-Score of E. The Nutri-Score was displayed on 75 products (10.2%). 

### 3.4. NOVA Classification

In our sample (*n* = 735), 39 products (5.3%) were NOVA 1, 21 products (2.9%) were NOVA 3, and 675 products (91.8%) were NOVA 4 (UPFs). No products were NOVA 2. For products classified as NOVA 4 (*n* = 675), 652 products (92.9%) contained ultra-processing markers, 627 products (92.9%) contained additives, 622 products (92.4%) had 5 or more ingredients, and 146 products (21.6%) resulted from a series of industrial processes (e.g., extrusion, cracking, etc.). 

### 3.5. Compliance with the WHO Nutrient Profile Model

Most products did not meet the criteria of the WHO NPM (*n* = 682, 92.8%). For these products, thresholds were exceeded for added sugars (*n* = 490, 71.8%), sodium (*n* = 212, 31.1%), total fat (*n* = 167, 24.4%), total sugars (*n* = 80, 11.7%), non-sugar sweeteners (*n* = 55, 8.1%), saturated fat (*n* = 21, 3.1%), and energy (*n* = 2, 0.3%). 

### 3.6. Nutri-Score and NOVA Classification

The less processed products generally had a better Nutri-Score (*p* < 0.001). The majority of products in the NOVA 1 category had a Nutri-Score of A (*n* = 19, 48.7%). Most of the products in the NOVA 3 and NOVA 4 categories had a Nutri-Score of C (*n* = 11, 52.4%) and D (*n* = 279, 41.3%), respectively. Regarding the distribution of the NOVA categories according to the Nutri-Score, all products with a Nutri-Score of E (*n* = 138, 100%) were NOVA 4. The detailed distribution is presented in Figure 2. 

### 3.7. Nutri-Score and WHO NPM

The WHO NPM categories with the most products with a Nutri-Score of A were fresh or dried pasta, rice and grains (*n* = 4, 66.7%), processed fruit and vegetables (*n* = 12, 60.0%), and savory plant-based foods/meat analogs (*n* = 6, 30.0%). Conversely, the categories with the most products with a Nutri-Score of E were energy drinks (*n* = 15, 79.0%); plant-based milk (*n* = 6, 60.0%); and cakes, sweet biscuits, and pastries (*n* = 42, 45.2%) (Appendix A). Products that meet the NPM criteria generally had a better Nutri-Score (*p* < 0.001). The compliance rate with the WHO NPM for each Nutri-Score category is presented in Figure 3. 

### 3.8. WHO NPM and NOVA Classification

For products that met the WHO NPM criteria (*n* = 53), 16 products (30.2%) were NOVA 1, 3 products (5.7%) were NOVA 3, and 34 products (64.2%) were NOVA 4. For products that did not meet the WHO NPM criteria (*n* = 682), 23 products (3.4%) were NOVA 1, 18 products (2.6%) were NOVA 3, and 641 products (94.0%) were NOVA 4. More than two-thirds of the WHO NPM food groups (15/22) contained 75.0% or more products belonging to the NOVA 4 category. Conversely, one food group (juices) had 75.0% of unprocessed products (NOVA 1). Products that meet the WHO NPM criteria were generally less processed (*p* < 0.001). 

### 3.9. Organic Products

A total of 102 products (13.9%) of the sample were organic. Concerning compliance with the WHO NPM, 22 out of 102 organic products (21.6%) complied with the nutritional criteria, compared with 31 out of 633 non-organic products (4.9%). Organic products generally had a better Nutri-Score, were less processed, and met the NPM criteria more often (p_s_ < 0.001). The distribution of Nutri-Score categories by organic and non-organic categories in food and beverages marketed to children is presented in Figure 4. 

### 3.10. Nutritional Claims

A total of 290 products (39.5%) had one or more nutrition claim(s) on the packaging. The most common nutrition claims were related to vitamins (*n* = 76, 26.2%), reduced/less sugars (*n* = 84, 29%), and minerals (e.g., calcium, iron) (*n* = 55, 19.0%). Products with nutrition claims generally had a better Nutri-Score (*p* < 0.01) and were less processed (*p* < 0.03). Nutrition claims and compliance with the WHO NPM were not statistically associated (*p* = 0.37).

## 4. Discussion

This is the first study to assess the nutritional quality of pre-packed foods marketed to children available on the Swiss market. A total of 735 products, selected from the two largest supermarkets, two hard discounters, and one organic food grocery store, were analyzed. The most common WHO NPM categories were (1) confectionaries and desserts; (2) savory snacks; and (3) cakes, sweet biscuits, and pastries. The most common marketing techniques used on over 30% of products were childish names/fonts, special characters, and/or children’s drawings. More than half of the products had a Nutri-Score of D or E. Over 90% of products were UPFs (NOVA 4). A high proportion (93%) of products did not meet the criteria of the WHO NPM. Food and beverages marketed to children assessed on the Swiss market were, therefore, mainly of poor nutritional quality. Organic food and products with a nutrition claim generally had a better Nutri-Score and were less processed. Of note, there are few studies analyzing the nutritional quality of pre-packaged products marketed to children using the same methodology.

In our study, the most common WHO NPM categories were confectionaries and desserts (25%); savory snacks (17%); and cakes, sweet biscuits, and pastries (13%). These findings regarding sugary products are consistent with other international studies [13,24]. In fact, sugary products are generally the most represented among pre-packaged products marketed to children, such as cakes, sweet biscuits, and pastries [24]; breakfast cereals [13,24]; or ice creams [13]. In our study, savory snacks were present to a higher extent compared to other studies [13,24] because, at the time of data collection, there was a contest targeting adolescents advertised on the packaging of several potato chip products, which made them eligible for study inclusion. Without the contest advertised on the packaging, those products would not have been included.

The most common marketing techniques found in this study were also those identified in other studies worldwide [11,32,34,44,45]. The use of special characters (mainly created by the brand) on more than one-third of products shows the brands’ intentions to create a link with children by developing their own mascot. In addition, the fact that up to six different marketing techniques per product were found shows the intensive marketing of the food industry to attract children’s attention and strengthen their appeal for these products [7,10,46]. These marketing techniques influence children’s food choices and preferences [10]. If these highly marketed products are mainly of poor nutritional quality, as shown in our study, and consumed regularly, they can have a negative impact on children’s health [10,17,22].

The proportion of products marketed to children displaying the Nutri-Score label was only 10% in the present study, while Richonnet et al. found a proportion of 21% in a recent French study dedicated to pre-packed foods marketed to children [24]. This could be explained by the fact that the Nutri-Score has been introduced in France for a longer time than in Switzerland (2017 vs. 2019) and is better known [27,47,48]. However, the Nutri-Score is displayed on 50% of products in the French market (2020), suggesting that children’s products display the Nutri-Score less often on their packaging than general products.

In our study, 58% of products had a Nutri-Score of D or E (lower nutritional quality), which is close to the proportion found in foods sold to French children (59%) [24]. In France, a study analyzing the general food market found that 31% of all foods sold in supermarkets, not only those for children, had a Nutri-Score of D or E [47]. Such data do not exist for the Swiss market but suggest that children’s products are generally of poorer nutritional quality than the rest of the product range. The consumption of well-graded products is associated with better diet quality [49,50,51], whereas Nutri-Score D and E products are often HFSS products and UPFs. This can adversely affect children’s health if eaten in the proportion exceeding nutritional recommendations [52]. A cohort study conducted in 10 European countries and including 501,594 adults showed that the consumption of products with a poorer Nutri-Score was associated with a higher risk of mortality [53]. Other cohort studies among adults showed an association between the consumption of products with low Nutri-Scores and weight gain, a higher BMI [54], and an increased risk of metabolic syndrome [55].

Most of the products were UPFs or NOVA 4 (92%) and only 5% were unprocessed or minimally processed (NOVA 1). Similar results were found in France for food marketed to children (88% NOVA 4 and 7% NOVA 1) [24]. Several studies using the NOVA classification have also found high proportions of UPFs among pre-packaged food (for adults and children): 64% in China [56], 71% in the United States [57], and 83% in New Zealand [58]. The significant presence of UPFs, particularly among children’s pre-packed products, is consistent with the fact that UPF consumption is higher among children than adults. For example, the level of UPF consumption represented 67% of total energy intake in the United States [59], 66% in the United Kingdom (UK) [60], 46% in France [61], and 33% in Belgium [62] among children, compared with, respectively, 57% [63], 54% [64], 36% [65], and 30% [62] among adults. Many studies have established an association between high consumption of UPFs and an increased risk of being overweight or obese [8,9,66,67], having metabolic syndrome [8,9], and experiencing dyslipidemia [8,68]. In addition, an inverse relationship was also found between the level of UPF consumption and the overall diet quality [69,70]. These findings led the European Childhood Obesity Group to call for raising awareness among children and adolescents about the negative impacts of UPF consumption and for implementing public health measures aiming at reducing their consumption, such as restricting marketing of UPFs to children and taxing or regulating the sale of UPFs around schools [71].

The majority of products (93%) did not meet the criteria of the WHO NPM. The few studies assessing this aspect also found high rates of non-compliance in products marketed to children: 90% (WHO NPM 2023 version) [72] and 95% (2015 version) [24] in France, and 93% in Slovenia (version 2015) [13]. This large number of products should not be marketed according to the WHO because of their poor nutritional quality. Foodwatch, a European non-profit consumer rights organization, also showed in a recent study that 96% of the analyzed products in France did not meet the criteria of the WHO NPM, even though these products were marketed by companies who committed to the EU Pledge [72]. This underscores the disparity between the intended goals and the actual outcomes within the children’s food market.

Less processed products (NOVA 1) generally had a better Nutri-Score, as found in France [24], even though about 90% of Nutri-Score B products were UPFs in France and Switzerland. Two other French studies that did not focus only on children’s products showed that the poorer the Nutri-Score, the higher the proportion of UPFs [73,74]. The products that met the WHO NPM criteria generally had a better Nutri-Score and mainly had a Nutri-Score of A or B (respectively, 43% and 40%). Comparable proportions were found in food marketed to children on the French market [24]. These results demonstrate the consistency between the WHO NPM and the Nutri-Score. Logically, categories with HFSS products had the highest proportions of products with a Nutri-Score of E. Surprisingly, plant-based milk, which has been found to be perceived by parents as being healthier than cow’s milk or for their children’s health [75,76], mainly had a Nutri-Score of E (60%) and non-compliant with the NPM (60%). The main reason is the high proportion of added sugars. The fact that more than half of breakfast cereals had a Nutri-Score of A or B is surprising given their high added-sugar content. The explanation is probably the addition of nutrients (e.g., fibers), which improve the Nutri-Score. However, the new algorithm of the Nutri-Score is stricter with some product categories, and breakfast cereals with high sugar content, for example, have a less favorable classification [38].

Our results suggest that the nutritional quality of organic products is generally better compared to non-organic products. However, the nutritional quality of children’s organic products is still poor. In fact, 78% of organic products did not meet the NPM criteria, and 75% were UPFs. These findings are comparable with French data, where 63% of products were non-compliant and 65% were UPFs [24]. In addition, almost half of the products (40%) had one or more nutrition claims. These results are similar to children’s food sold in France (36%) [24], the UK (42%) [11], Australia (56%) [44], Brazil (51%) [45], and, to a greater extent, Canada (63%) [34]. The nutrition claims are mainly related to sugar and micronutrients, as in the UK [11] or in France [24]. In our study, the proportions of products with a nutrition claim that had a Nutri-Score of A or B were low (respectively, 15% and 16%), and those that were UPFs and NPM non-compliant products were high (respectively, 92% and 90%). Therefore, the results contrast with the fact that products with a nutrition claim are healthier, as generally perceived by consumers (‘health halo’) [11]. Thus, the organic label or the presence of a nutrition claim is not a guarantee of minimally processed products and optimal nutrient composition for foods marketed to children.

This study focused on packaging, but marketing techniques target children through many other platforms and media such as advertising on TV, social media, mobile apps, or digital games. Despite consensus on the need for regulations to limit HFSS marketing to children, governments struggle to implement effective measures to limit this exposure. Voluntary guidelines and self-regulations for responsible food marketing to children have shown limited impact on the nutritional quality of foods marketed and on exposure [77]. Evidence on statutory regulation, however, has shown an encouraging reduction in children’s exposure to advertising for HFSS [78]. In the world, implementing legislation to restrict food marketing for children is often slowed down by political and technical challenges, but can be successful. The 2010 WHO resolution WHA63.14, the recent WHO guideline on policies to protect children, and several published resources support this goal [79,80]. For example, in Norway, the government banned advertising of unhealthy food and beverages to children under 18 using the WHO NPM [81]. Advocacy groups are also in favor of strong actions. The European Consumer Organisation has called for stricter measures to protect children from food marketing, such as the ban of marketing for unhealthy products or the removal of marketing on packaging [82]. Another example is the European Best-ReMap project aiming to reduce the impact of harmful marketing of food to children by bringing together health professionals and stakeholders to discuss concrete solutions to achieve this goal [83]. Regarding food manufacturers, front-of-package nutrition labels, such as a Nutri-Score, can promote product reformulation towards healthier recipes and, therefore, improved nutritional quality [84,85].

One limitation of our study is the reliance on only one registered dietitian to collect the data in the food stores, so there may have been subjectivity in the inclusion/exclusion of products. The lack of systematic control by another independent researcher for data entry into the database is another limit. However, a standardized and piloted protocol was used and the quality control on 10 products with another senior dietitian showed high agreement between the two dietitians. In addition, the lack of information on a small proportion of products led to the estimation of fiber content and the proportion of fruits, vegetables, nuts, legumes, or oils. These factors may have minimally influenced the calculation of the Nutri-Score. The fact that the NOVA classification is not standardized may have led to a subjective classification of some products. Despite the audit of the main Swiss food stores, we did not audit all the places where children’s food is sold (e.g., other supermarkets, convenience stores, markets, farms, etc.) due to resource and time constraints. Some products marketed to children were, therefore, not included. By design, we excluded unpackaged food and assumed nutritional values displayed on the packaging to be accurate. In addition, this study investigates only the food supply and not the children’s food consumption. Since there is currently no data on children’s food intake in Switzerland, it is not possible to know the extent to which these products are bought and eaten by children. Finally, we did not perform analyses by food stores because identical products were not photographed several times.

Despite these limitations, our study has a number of strengths. Firstly, the data were collected from the main Swiss food stores to obtain a large and representative sample of the food products sold on the market. Secondly, the data collection was carried out in a standardized way using a standardized and piloted protocol. Thirdly, the product classification was based on validated tools (Nutri-Score [27], NOVA [28], and WHO NPM [29]) that took several dimensions of nutritional quality into account. Finally, the use of these three tools makes the results of this work comparable with other international studies that have used these classifications.

## 5. Conclusions

Our study shows that pre-packed food and beverages marketed to children assessed on the Swiss market were mainly of poor nutritional quality. Most products should not be marketed according to the WHO and were UPFs. The industry self-regulation to restrict unhealthy food marketing is of limited effect in Switzerland. Public health measures should be adopted to improve the nutritional quality of foods marketed to children and restrict the marketing of unhealthy foods to children in Switzerland.

## Figures and Tables

**Figure 1 nutrients-16-01656-f001:**
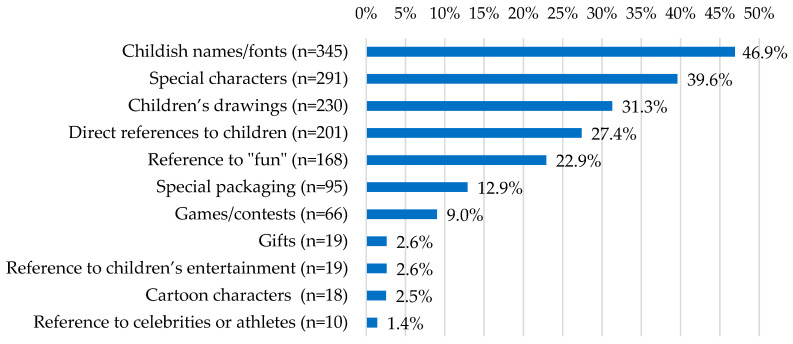
Proportions of marketing techniques found on the packaging of food and beverages marketed to children.

**Figure 2 nutrients-16-01656-f002:**
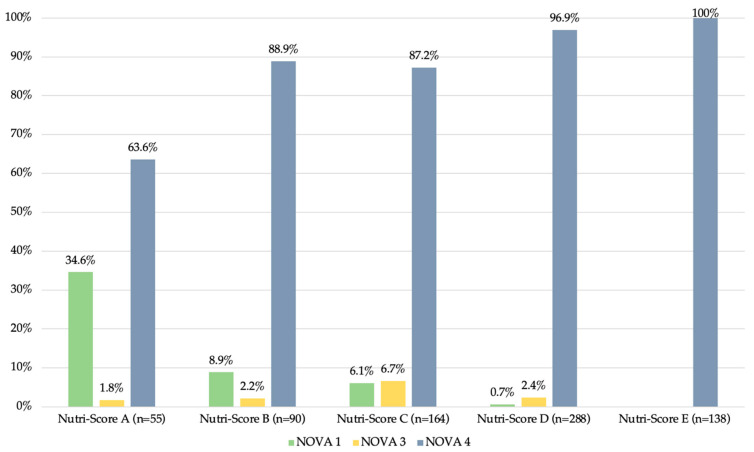
Proportions of the NOVA categories by Nutri-Score categories in food and beverages marketed to children (*n* = 735).

**Figure 3 nutrients-16-01656-f003:**
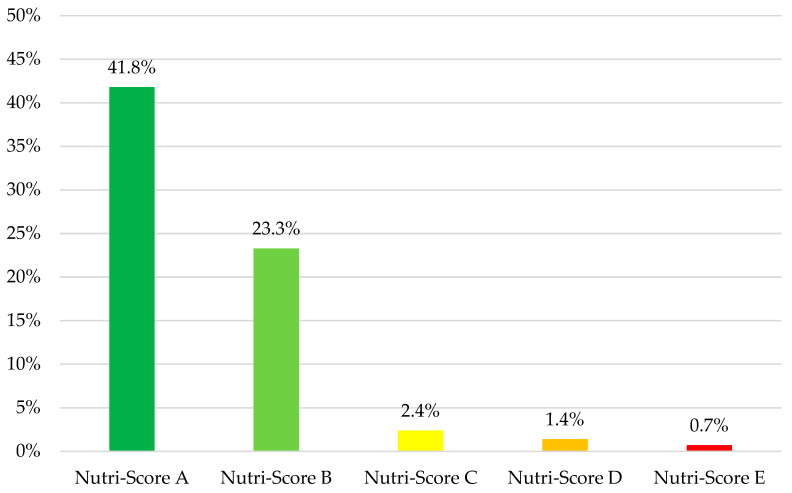
Rate of compliance with the WHO NPM by Nutri-Score categories in food and beverages marketed to children.

**Figure 4 nutrients-16-01656-f004:**
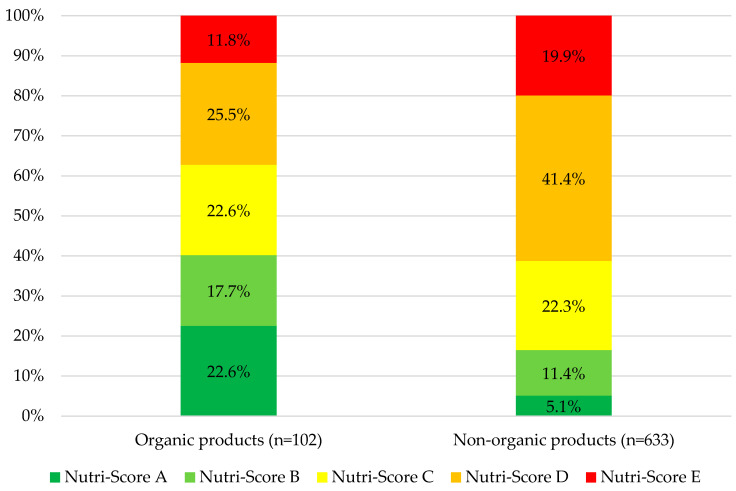
Distribution of Nutri-Score categories by organic and non-organic categories in food and beverages marketed to children.

## Data Availability

The raw data supporting the conclusions of this article will be made available by the authors upon request due to privacy.

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
