# Peer review of "What Is the Nutritional Quality of Pre-Packed Foods Marketed to Children in Food Stores? A Survey in Switzerland"

_nutrients, 2024, doi:10.3390/nu16111656_

Round 1

Reviewer 1 Report

Comments and Suggestions for Authors

The research is very exciting and important. Maybe the manuscript is a bit long. I recommend shortening the Introduction section, especially its last paragraph. The information that appears in this paragraph is repeated in the Materials and Methods chapter. I also recommend reducing the Discussion section. Nevertheless, this is a very valuable publication that will be useful to many people.

Reviewer 2 Report

Comments and Suggestions for Authors

The manuscript “What is the nutritional quality of foods marketed to children in food stores? A survey in Switzerland” is an interesting work with nice information may only concerns are the following:

In section 2.1 The selection of products was done by a registered dietitian who visited 5 stores. This work was done in 10 days. I think it is a big work to go to 5 supermarkets to revise all foods and beverages targeting children age 3-18 years. How was it done, did you ask for permission in the supermarkets as they usually do not allow buyers to take photos of the products, at least in my country.  Additionally, I think that it is a lot of work for only one person, how was it done? How the information was collected?

I think that it is better to put Definition of nutritional quality before of section 2.2 ( Data entry)

There are some theorical explanations given in methodology that maybe it will be better described in introduction section, in order to simplify methodology section. Lines130 till 135 should better be included in introduction section. The same with lines 155-159

Line 165. NOVA4 products were made using highly processed industrial techniques … How do you know? In the packages there is not information on the processing techniques employed.

Line 296-“Organic products generally had a better Nutri-Score, were less processed, and met the NPM criteria more often”. I think that maybe this phrase is confusing as organic products are usually fresh foods. I do not know if it is correct to say better Nutri-core. They must be compared with homologues ones.

Reviewer 3 Report

Comments and Suggestions for Authors

In an  Europe that missed its target, to stop the growing of the number of obese children until 2020, marketing is an important tool. Recent joint actions aimed to evaluate what can be done, especially in a digital era where unhealthy products are marketed not only in shops or at the television, but also online. Pledges with industry seem not to work so a more serious way to address the problem is needed. The authors evaluate exactly this, the nutritional value of foods  from Swiss  targeted to children. They use 11 "hints" that show that a product is "for children" and analyze its nutrients  taking as markers Nutriscore, NOVA classification for food processing  and the WHO NPM.  Every step of the work is clearly presented, so that even readers not familiarized with the above classifications can understand what has been done. Being organic  and having claims are other 2 aspects that are taken into consideration. 

The results of the study are unfortunately not good for our children, with a overwhelming number of products having a low nutriscore, being highly processed and not complying with WHO criteria. Even more, many products don`t even use nutriscore (which is not against the laws but which presence could be a helper in selecting a better product, if someone wants to do it). 

Figures are interesting and clear, the presentation is good and the tools used are adequate. Maybe in the discussion chapter some more elements could be added, comparing the findings of the study with similar studies carried elsewhere, as well as reminding some of the European projects that recently covered exactly this topic (like ReMap, for example)

Advertising to children is not adequately regulated in Europe, this sector has to be clearly regulated, just pledges with industry seem not to work. Maybe you can add a little more about what has been studied in this area elsewhere in Europe, as well as some projects carried out and their results. 
